# A Novel Prognostic Prediction Model Based on Pyroptosis-Related Clusters for Breast Cancer

**DOI:** 10.3390/jpm13010069

**Published:** 2022-12-28

**Authors:** Baoxing Tian, Kai Yin, Xia Qiu, Haidong Sun, Ji Zhao, Yibao Du, Yifan Gu, Xingyun Wang, Jie Wang

**Affiliations:** 1Department of Breast Surgery, Tongren Hospital, Shanghai Jiao Tong University School of Medicine, No. 1111, Xianxia Road, Shanghai 200336, China; 2Hongqiao International Institute of Medicine, Tongren Hospital, Shanghai Jiao Tong University School of Medicine, No. 1111, Xianxia Road, Shanghai 200336, China

**Keywords:** breast cancer, prognostic signature, pyroptosis clusters, biomarker, TCGA

## Abstract

Breast cancer (BC) is the most common cancer affecting women and the leading cause of cancer-related deaths worldwide. Compelling evidence indicates that pyroptosis is inextricably involved in the development of cancer and may activate tumor-specific immunity and/or enhance the effectiveness of existing therapies. We constructed a novel prognostic prediction model for BC, based on pyroptosis-related clusters, according to RNA-seq and clinical data downloaded from TCGA. The proportions of tumor-infiltrating immune cells differed significantly in the two pyroptosis clusters, which were determined according to 38 pyroptosis-related genes, and the immune-related pathways were activated according to GO and KEGG enrichment analysis. A 56-gene signature, constructed using univariate and multivariate Cox regression, was significantly associated with progression-free interval (PFI), disease-specific survival (DSS), and overall survival (OS) of patients with BC. Cox analysis revealed that the signature was significantly associated with the PFI and DSS of patients with BC. The signature could efficiently distinguish high- and low-risk patients and exhibited high sensitivity and specificity when predicting the prognosis of patients using KM and ROC analysis. Combined with clinical risk, patients in both the gene and clinical low-risk subgroup who received adjuvant chemotherapy had a significantly lower incidence of the clinical event than those who did not. This study presents a novel 56-gene prognostic signature significantly associated with PFI, DSS, and OS in patients with BC, which, combined with the TNM stage, might be a potential therapeutic strategy for individualized clinical decision-making.

## 1. Introduction

Female breast cancer (BC) has become the most commonly diagnosed cancer, with an estimated 2.3 million new cases, and is the fifth leading cause of cancer mortality worldwide, with 685,000 deaths occurring in 2020 [1]. Independently of histological subtypes, molecular subtypes were categorized as Luminal-like (including Luminal A and Luminal B), HER2-positive, and triple-negative breast cancer (TNBC). These were based on tumor hormonal status, human epidermal growth factor receptor 2 (HER2) status, and proliferation marker protein Ki-67 (MKI67) status [2,3]. Based on TNM staging, breast cancer was categorized into five different stages (0, I II, III, IV) [4]. In the course of clinical personalized treatment, patients are further subdivided into advanced-stage (stages III and IV) and early-stage (stages 0, I and II) BC [5,6]. Molecular subtype and clinical staging could assist in deciding on therapeutic options for patients with BC [7], for instance, endocrine therapy for Luminal-like BC [8,9], HER2-targeted therapy for HER2-positive BC [10,11], and chemotherapy for TNBC [12,13]. Moreover, some authors showed that the ADH/ALDH activities are lower in tumor cells than in normal parenchyma, suggesting that isoenzymes of ADH may play an important role in carcinogenesis. Additionally, among all tested classes of ADH isoenzymes, only class I had higher activity in the serum of patients with breast cancer in stage IV [14,15]. However, it is still difficult to decide on the most appropriate systemic therapy or combination of therapies for a patient with BC based on only the traditional clinicopathological prognostic factors [16].

With the advent of the era of big data and precision medicine, a variety of multigene prognostic tools, including Oncotype Dx and MammaPrint, have been established to predict outcomes and aid in adjunct therapy decision-making in patients with ER-positive, HER2-negative BC that are either lymph-node-negative or node-positive (one to three metastatic nodes) [17,18]. However, none can predict prognosis in all types of BC, and the prognosis for some patients with BC is still not good. Additionally, due to the disease heterogeneity, the variation in treatment responses among patients, and the lack of targeted therapy, there is an urgent need to improve the therapeutic strategy against BC.

Considering the significance of the host immune system in controlling the progression and spread of solid tumors, immunotherapy in breast cancer is currently being evaluated as a new arm of treatment, in combination with existing therapeutic strategies [19]. Recently, immunotherapy for BC has received more attention and has increasingly been used in clinical practice. As of 15 September 2020, 82 trials were identified on Clinicaltrials.gov and EURODACT as investigating immunotherapy for early BC [20]. The efficacy of immunotherapy is largely determined by the complex immune microenvironment of the tumor [21,22]. A complex tumor microenvironment (TME) consisting of immune cells (regulatory T cells, macrophages, B cells, etc.) and stromal cells (fibroblasts, endothelial cells, and adipocytes) allows for overt immune escape and tumor progression to occur. Pyroptosis, an inflammatory type of programmed cell death, accompanied by inflammatory and immune responses, can influence the proliferation, invasion, and metastasis of tumors [23,24,25,26]. Thus, it is imperative to find an effective pyroptosis-related signature for the prediction of BC prognosis.

Herein, we took advantage of RNA-Seq and clinical datasets from TCGA to construct a novel prognosis prediction signature for the progression-free interval (PFI) based on pyroptosis-related clusters, disease-specific survival (DSS), and overall survival (OS) in patients with BC, which not only provided important insights into the molecular and signaling pathways of pyroptosis but also provided a theoretical reference that is significant for individualized clinical decisions for patients with BC.

## 2. Methods

### 2.1. RNA-Seq and Clinical Datasets

Level 3 RNA-seq data (tumor, *N* = 1110 and normal, *N* = 113) and accompanying clinical datasets were obtained through the TCGA Genomic Data Commons portal (GDC, https://cancergenome.nih.gov/ (Data Release 31.0—29 October 2021)). Differential pyroptosis-related gene expression analysis was achieved using RNA-seq data. Then, samples from 1025 female patients with BC, cataloged by TCGA, for which clinical features and RNA-seq data were available, were selected for further analyses. The BC dataset used, from TGCA, was last updated on 29 October 2021. The presented results are completely based on data generated by TCGA Research Network.

### 2.2. Identification of Pyroptosis-Related Clusters

Strawberry Perl software was used to merge RNA-seq files, and the Normalize Quantiles (edgeR, R package) was used to normalize RNA expression levels. A total of 51 pyroptosis-related genes were obtained from the Reactome website (https://reactome.org/ (accessed on 31 October 2021)), and 38 differential pyroptosis-related genes were determined using the R package “limma” (FDR ≤ 0.05). According to the 38 differential pyroptosis-related genes, pyroptosis-related clusters were built by the R package “ConsensusClusterPlus (version 1.50.0, Wilkerson MD, Hayes DN. Lineberger Comprehensive Cancer Center, University of North Carolina at Chapel Hill, Chapel Hill, NC 27599, USA)”.

### 2.3. Bioinformatic Analysis of the Two Pyroptosis-Related Clusters

A total of 434 DEGs associated with the two pyroptosis-related clusters were identified. GO functional enrichment and KEGG pathway analysis were performed to discover biological processes overrepresented in the 434 gene list by R package “clusterProfiler”, “org.Hs.eg.db”, “enrichplot”, “ggplot2”, and “graphlayouts”. To identify potential interactions between DEGs, a protein–protein interaction (PPI) network was analyzed using the String database (http://string-db.org/ (accessed on 12 November 2021)).

### 2.4. Risk Prognostic Model Construction and Evaluation

The dataset (*N* = 1025) was used to develop a clinical prognostic prediction model, and half of the dataset (*N* = 512) was randomly selected as a validation dataset, which was used to verify the accuracy of the RNA-based prognostic model as a predictor of PFI of patients with BC. Univariate and multivariate Cox proportional hazard models were constructed as risk prognostic models, and area under the curve (AUC) analysis was conducted to evaluate their accuracy.

### 2.5. Nomogram Construction

The independent predictors, evaluated by univariate and multivariate Cox regression, were included to establish a nomogram model by the R package “rms”, “foreign”, and “survival”, aiming to evaluate the predictive power of independent predictors for 5-year, and 10-year PFI and DSS rates. Subsequently, a calibration plot was established to evaluate the accuracy of the nomogram prediction with the R package “rms” and “survival”.

### 2.6. Statistical Analyses

Statistical analyses were conducted using SPSS statistical suite version 25.0 (IBM *SPSS* Statistics, Chicago, USA), Strawberry Perl version 5.32.1.1 (https://strawberryperl.com/release-notes/5.32.1.1-32bit.html (accessed on 12 November 2021)), and R version 3.6.1 (R Project for Statistical Computing, https://www.r-project.org (accessed on 12 November 2021)). Statistical significance was defined as a two-sided *p*-value or adjusted *p*-value ≤ 0.05. The primary outcome in this study was PFI, the secondary outcome was DSS, and the event times of PFI, DSS, and OS were defined according to the guidelines for time-to-event endpoint definitions in the BC trial [27]. The PFI, DSS, and OS event times for the individual patients enrolled in this retrospective study were manually retrieved from TCGA clinical records and a previous study [28,29]. The Wilcoxon test (FDR ≤ 0.05) was used to identify the DEGs for variables of two groups using the R package “limma”. The DEGs with prognostic value, identified using univariate Cox proportional hazard models (*p*-value ≤ 0.05), were further analyzed by multivariate Cox regression (default settings: backward, conditional, entry 0.05, removal 0.10), and 56 pyroptosis-related DEGs identified by this analysis were used to construct a formula for the calculation of prognostic risk scores. The results of univariate and multivariate Cox regression are presented as a hazard ratio (HR), with a 95% confidence interval (CI). The cut-off point was calculated by the R package “ggrisk”. Next, BC cases were categorized as “low risk” or “high risk” based on risk scores being higher or lower than the cut-off point. Chi-square analysis was used to assess the correlation between BC clinicopathological features and risk subgroups. Kaplan–Meier (KM) analyses were applied to generate survival curves and the log-rank test was used to establish the significance of differences between curves. The receiver operating characteristic (ROC) curve was constructed to assess the prognostic performance of the risk score. A prognostic nomogram to predict individual survival based on the signature and clinical risk factors was constructed by Cox regression. The accuracy of the risk prognostic model was tested using AUC (95% CI) values.

## 3. Results

### 3.1. Identification of Pyroptosis-Related Clusters

A total of 38 differential pyroptosis-related genes were determined between normal and tumor samples (normal, *N* = 113 and tumor, *N* = 1110) from 51 pyroptosis-related genes, and the expression levels of the 38 genes were shown using the R package “pheatmap” (Figure 1A). Subsequently, based on the 38 differential pyroptosis-related genes, two pyroptosis-related clusters (cluster C1, *N* = 439; cluster C2, *N* = 586) were built using the R package “ConsensusClusterPlus” (Figure 1B–D), and principal components analysis (PCA) and t-Distributed Stochastic Neighbor Embedding analysis (tSNE) plots were constructed to show the two clusters with obvious alterations using the R package “Rtsne” (Figure 1E,F). KM analysis indicated that patients in the low-risk group had significantly longer PFI, DSS, and OS (PFI, *p* = 0.001, DSS, *p* = 7 × 10^−4^, OS, *p* < 0.0001) (Figure 1G–I).

### 3.2. Identification of DEGs and Bioinformatic Analysis of the Two Pyroptosis-Related Clusters

Differential gene expression analysis was run on the two pyroptosis-related clusters using the Wilcoxon test. A total of 434 DEGs were identified and are shown in the pheatmap (Figure 2A). GO functional enrichment (Figure 2B) and KEGG pathway (Figure 2C) analysis demonstrated that the DEGs were enriched in immune-related pathways, such as humoral immune response, adaptive immune response based on the somatic recombination of immune, B-cell-mediated immunity, complement activation, and immunoglobulin-mediated immune response. To further estimate the level of individual tumor-infiltrating immune cells in the two pyroptosis-related clusters, we performed CIBERSORT, and the landscape of tumor-infiltrating immune cells is shown in the barplot (Figure 2D). Some types of tumor-infiltrating immune cells were variously distributed in different pyroptosis-related clusters, such as naïve B cells, plasma cells, T cells CD8, T cells CD4 memory resting, T cells CD4 memory-activated, T cells follicular helper, T cells gamma delta, NK cells resting, macrophages M0, macrophages M1, macrophages M2, dendritic cells resting, mast cells resting, mast cells activated, and neutrophils (Figure 2E). Subsequently, the PPI was constructed to visualize the interactions among 434 DEGs (Figure 2F).

### 3.3. Construction of Risk Model Based on Pyroptosis-Related DEGs

The univariate Cox proportional hazard analysis identified 256 DEGs with prognostic value, 56 of which were determined by multivariate Cox regression to be the optimum prognostic models for predicting PFI risk in patients with BC (Figure 3A). Risk scores were calculated using the formula construct according to multivariate Cox regression (Risk calculation formula). Based on the risk score, −0.956728925, which was calculated as the cut-off point, the patients were grouped into high- (*N* = 308) and low- (*N* = 717) risk groups. Patients with high-risk scores tended to present poorer clinical outcomes compared with patients with low-risk scores (Figure 3B). The expression levels of the 56 genes are shown in a violin plots (Figure 3C).

KM analysis indicated that patients in the low-risk group had significantly longer PFI, DSS, and OS in the training and validation datasets (all *p* < 0.0001) (Figure 3D–F,J–L). ROC curve analysis shows that the 56-gene signature had good sensitivity and specificity for predicting PFI, DSS, and OS in the training and validation datasets (training dataset, PFI, AUC = 0.768, 95% CI 0.727–0.810, *p* < 0.001; DSS, AUC = 0.744, 95% CI 0.689–0.800, *p* < 0.001; OS, AUC = 0.634, 95% CI 0.582–0.686, *p* < 0.001; validation dataset, PFI, AUC = 0.793, 95% CI 0.737–0.849, *p* < 0.001; DSS, AUC = 0.751, 95% CI 0.667–0.836, *p* < 0.001; OS, AUC = 0.666, 95% CI 0.588–0.744, *p* < 0.001) (Figure 3G–I,M–O).

### 3.4. Clinicopathological Features

A total of 1025 female cases with BC recorded in TCGA were extracted for analysis in this study. The median patient age was 58 years (ranging from 26 to 90 years), while the median PFI was 767 days, and DSS was 825 days. The 10-year PFI rate for all analyzed cases was 87.6%, and 10-year DSS was 92.9%. BC tumor size, lymph node, and metastasis status (TNM) stage were defined as outlined by the Eighth Edition American Joint Committee on Cancer (AJCC) Staging Manual [4], and molecular subtype (PAM50) was derived from a previous report by Thorsson et al. [28]. In the age subgroup, the proportion of ≥61 y subgroup patients in the high-risk group was significantly higher than that in the ≤40 y and 41–60 y subgroup in the training dataset (χ^2^ = 6.492, *p* = 0.040), but not in the validation dataset (χ^2^ = 5.661, *p* = 0.059). In the molecular subgroup, the proportion of luminal A subgroup patients in the high-risk group was significantly lower than that in the other subgroup in the training dataset (χ^2^ = 10.957, *p* = 0.027), but not in the validation dataset (χ^2^ = 6.174, *p* = 0.187). Further, metastasis status was associated with a higher proportion of patients in the high-risk group for both the total dataset (χ^2^ = 11.582, *p* = 0.001) and validation dataset (χ^2^ = 7.243, *p* = 0.011). The demographic and clinical, pathologic characteristics of the patients with breast cancer are shown in Table 1.

### 3.5. 56-Gene Signature Associated with Prognosis of Patients with BC

Univariate and multivariate Cox proportional hazard regression analyses for 10-year PFI indicated that a higher 56-gene risk score was correlated with a higher incidence of clinical events (univariate analysis, HR = 6.257, 95% CI: 4.331–9.039, *p* < 0.001; multivariate analysis, HR = 5.643, 95% CI 3.894–8.175, *p* < 0.001). Furthermore, univariate and multivariate Cox proportional hazard regression analyses for 10-year DSS also indicated that a higher 56-gene risk score was correlated with a higher incidence of clinical events (univariate analysis, HR = 5.520, 95% CI: 3.407–8.944, *p* < 0.001; multivariate analysis, HR = 4.578, 95% CI 2.797–7.494, *p* < 0.001). The results of univariate and multivariate Cox proportional hazard regression analyses for 10-year PFI and DSS are shown in Table 2.

Furthermore, KM survival curves for 10-year PFI, DSS, and OS showed that the high-risk group had a worse prognosis in both the training (all, *p* < 0.0001) and validation (all, *p* < 0.0001) datasets (Figure 3D–F,J–L). To determine the sensitivity and specificity of the prognostic signature for predicting survival, we conducted ROC analyses of the training and validation datasets. ROC curves showed that the prognostic signature had good sensitivity and specificity for predicting survival for 10-year PFI, DSS, and OS in both the training (PFI, AUC = 0.768, 95% CI 0.727–0.810, *p* < 0.001; DSS, AUC = 0.744, 95% CI 0.689–0.800, *p* < 0.001; OS, AUC = 0.634, 95% CI 0.582–0.686, *p* < 0.001) and validation (PFI, AUC = 0.793, 95% CI 0.737–0.849, *p* < 0.001; DSS, AUC = 0.751, 95% CI 0.667–0.836, *p* < 0.001; OS, AUC = 0.666, 95% CI 0.588–0.744, *p* < 0.001) datasets (Figure 3G–I,M–O).

### 3.6. Evaluation of the Predictive Power of the Prognostic Signature

According to the AJCC cancer staging manual (eighth edition), the TNM stage is correlated with cancer prognosis [4,16,30]. Furthermore, age and intrinsic molecular subtype (PAM50) are closely linked to prognosis in patients with BC [31,32,33,34]. Furthermore, to validate the potential of the prognostic signature as a predictor of the PFI, DSS, and OS of patients with BC, the entire TCGA BC dataset was stratified by TNM stage, age, and molecular subtype. Patients were split into three age subgroups (≤40, 41–60, and ≥61 years old), three lymph node status subgroups (N0, N1, and N2–N3), three tumor size subgroups (T1, T2, and T3–T4), and five molecular subtype subgroups (PAM50, luminal A, luminal B, HER2, basal-like, and normal-like).

KM analysis indicated that patients in the low-risk group had significantly longer PFI, DSS, and OS in all three age subgroups (PFI, all subgroup, *p* < 0.0001; DSS, ≤40 y subgroup, *p* = 0.0082, 41–61 y subgroup, *p* < 0.0001, ≥61 y subgroup, *p* < 0.0001; OS, ≤40 y subgroup, *p* = 0.0038, 41–61 y subgroup, *p* < 0.0001, ≥61 y subgroup, *p* = 0.00057) (Figure 4A–C,G–I; Appendix A). ROC curve analysis showed that the prognostic signature had good sensitivity and specificity for predicting PFI, DSS, and OS in all three age subgroups (PFI, ≤40 y subgroup, AUC = 0.796, 95% CI 0.693–0.899, *p* < 0.001; 41–61 y subgroup, AUC = 0.765, 95% CI 0.703–0.808, *p* < 0.001; ≥61 y subgroup, AUC = 0.781, 95% CI 0.715–0.847, *p* < 0.001; DSS, ≤40 y subgroup, AUC = 0.771, 95% CI 0.661–0.881, *p* = 0.002; 41–61 y subgroup, AUC = 0.718, 95% CI 0.619–0.817, *p* < 0.001; ≥61 y subgroup, AUC = 0.762, 95% CI 0.680–0.844, *p* < 0.001; OS, ≤40 y subgroup, AUC = 0.716, 95% CI 0.595–0.838, *p* = 0.007; 41–61 y subgroup, AUC = 0.636, 95% CI 0.543–0.728, *p* < 0.004; ≥61 y subgroup, AUC = 0.603, 95% CI 0.529–0.678, *p* < 0.006) (Figure 4D–F,J–L; Appendix A).

In the analyses of tumor size subgroups, KM curves also showed that patients in the low-risk group had a significantly better prognosis for PFI, DSS, and OS than those in the high-risk group (PFI, T1 subgroup, *p* < 0.0001, T2 subgroup, *p* < 0.0001, T3–T4 subgroup, *p* < 0.0001; DSS, T1 subgroup, *p* = 0.0026, T2 subgroup, *p* < 0.0001, T3–T4 subgroup, *p* < 0.0001; OS, T1 subgroup, *p* = 0.0095, T2 subgroup, *p* < 0.00012, T3–T4 subgroup, *p* < 0.0012) (Figure 5A–C,G–I; Appendix A). ROC analysis demonstrated that the prognostic signature had good sensitivity and specificity for predicting PFI, DSS, and OS in all three tumor size status subgroups (PFI, T1 subgroup, AUC = 0.756, 95% CI 0.661–0.850, *p* < 0.001, T2 subgroup, AUC = 0.758, 95% CI 0.700–0.816, *p* < 0.001, T3–T4 subgroup, AUC = 0.785, 95% CI 0.702–0.867, *p* < 0.001; DSS, T1 subgroup, AUC = 0.731, 95% CI 0.589–0.873, *p* = 0.005, T2 subgroup, AUC = 0.715, 95% CI 0.641–0.789, *p* < 0.001, T3–T4 subgroup, AUC = 0.789, 95% CI 0.684–0.894, *p* < 0.001; OS, T1 subgroup, AUC = 0.650, 95% CI 0.535–0.766, *p* = 0.017, T2 subgroup, AUC = 0.602, 95% CI 0.531–0.672, *p* < 0.006, T3–T4 subgroup, AUC = 0.667, 95% CI 0.561–0.772, *p* < 0.003) (Figure 5D–F,J–L; Appendix A).

In KM analyses, the curves showed that patients in the low-risk group had a significantly better prognosis for PFI, DSS, and OS than those in the high-risk group for all the lymph node subgroups (PFI, all the lymph node subgroup, *p* < 0.0001; DSS, N0 subgroup, *p* < 0.0001, N1 subgroup, *p* = 0.0027, N2–N3 subgroup, *p* < 0.0001; OS, N0 subgroup, *p* < 0.0001, N1 subgroup, *p* = 0.024, N2–N3 subgroup, *p* = 0.0022) (Figure 6A–C,K–M; Appendix A). ROC analysis demonstrated that the prognostic signature had good sensitivity and specificity for predicting PFI, DSS, and OS in all three lymph node status subgroups (PFI, N0 subgroup, AUC = 0.819, 95% CI 0.762–0.875, *p* < 0.001, N1 subgroup, AUC = 0.733, 95% CI 0.665–0.801, *p* < 0.001, N2–N3 subgroup, AUC = 0.764, 95% CI 0.676–0.853, *p* < 0.001; DSS, N0 subgroup, AUC = 0.805, 95% CI 0.717–0.894, *p* < 0.001, N1 subgroup, AUC = 0.694, 95% CI 0.613–0.775, *p* < 0.001, N2–N3 subgroup, AUC = 0.797, 95% CI 0.691–0.904, *p* < 0.001; OS, N0 subgroup, AUC = 0.638, 95% CI 0.544–0.733, *p* = 0.019, N1 subgroup, AUC = 0.599, 95% CI 0.520–0.678, *p* = 0.021, N2–N3 subgroup, AUC = 0.689, 95% CI 0.589–0.789, *p* = 0.001) (Figure 6F–H,J–L; Appendix A).

In the analyses of metastasis status subgroups, KM curves also showed that patients in the low-risk group had a significantly better prognosis for PFI than those in the high-risk group (M0 subgroup, *p* < 0.0001, M1 subgroup, *p* = 0.012) (Figure 6D,E), but not for DSS and OS (DSS, M0 subgroup, *p* < 0.0001, M1 subgroup, *p* = 0.2; OS, M0 subgroup, *p* < 0.0001, M1 subgroup, *p* = 0.36) (Figure 6N,O; Appendix A). ROC analysis demonstrated that the signature had good sensitivity and specificity for predicting PFI, DSS, and OS in M0 subgroups (PFI, AUC = 0.764, 95% CI 0.721–0.808, *p* < 0.001; DSS, AUC = 0.737, 95% CI 0.677–0.796, *p* < 0.001; OS, AUC = 0.620, 95% CI 0.565–0.674, *p* < 0.001), but not in the M1 subgroup (PFI, AUC = 0.385, 95% CI 0.014–0.755, *p* = 0.545; DSS, AUC = 0.573, 95% CI 0.221–0.853, *p* = 0.814; OS, AUC = 0.473, 95% CI 0.152–0.794, *p* = 0.865) (Figure 6I,J,S,T; Appendix A).

In the analyses of the five molecular subtype subgroups, KM curves also showed that patients in the low-risk group had significantly better prognosis for PFI and DSS than those in the high-risk group (PFI, normal-like subgroup, *p* < 0.0001, Luminal A subgroup, *p* < 0.001, Luminal B subgroup, *p* = 0.0006, HER2 subgroup, *p* = 0.0003, Basal-like subgroup, *p* < 0.0001; DSS, normal-like subgroup, *p* = 0.025, Luminal A subgroup, *p* < 0.0001, Luminal B subgroup, *p* = 0.0007, HER2 subgroup, *p* = 0.0057, Basal-like subgroup, *p* < 0.0001) (Figure 7A–E,K–O), but not all for OS (normal-like subgroup, *p* = 0.019, Luminal A subgroup, *p* = 0.0022, Luminal B subgroup, *p* = 0.064, HER2 subgroup, *p* = 0.1, Basal-like subgroup, *p* < 0.0001) (Appendix A). ROC analysis demonstrated that the signature had good sensitivity and specificity for predicting PFI and DSS in all five molecular subtype subgroups (PFI, normal-like subgroup, AUC = 0.783, 95% CI 0.689–0.876, *p* < 0.001, Luminal A subgroup, AUC = 0.744, 95% CI 0.673–0.815, *p* < 0.001, Luminal B subgroup, AUC = 0.692, 95% CI 0.561–0.824, *p* = 0.011, HER2 subgroup, AUC = 0.854, 95% CI 0.753–0.955, *p* = 0.001, Basal-like subgroup, AUC = 0.813, 95% CI 0.738–0.888, *p* < 0.001; DSS, normal-like subgroup, AUC = 0.735, 95% CI 0.618–0.853, *p* = 0.002, Luminal A subgroup, AUC = 0.698, 95% CI 0.594–0.802, *p* = 0.001, Luminal B subgroup, AUC = 0.751, 95% CI 0.611–0.891, *p* = 0.008, HER2 subgroup, AUC = 0.851, 95% CI 0.712–0.989, *p* = 0.009, Basal-like subgroup, AUC = 0.789, 95% CI 0.684–0.894, *p* < 0.001) (Figure 7F–J,P–T), but not all for OS (normal-like subgroup, AUC = 0.660, 95% CI 0.559–0.760, *p* = 0.004, Luminal A subgroup, AUC = 0.603, 95% CI 0.516–0.691, *p* = 0.028, Luminal B subgroup, AUC = 0.583, 95% CI 0.444–0.723, *p* = 0.247, HER2 subgroup, AUC = 0.635, 95% CI 0.411–0.859, *p* = 0.216, Basal-like subgroup, AUC = 0.769, 95% CI 0.661–0.878, *p* < 0.001) (Appendix A). These KM and ROC curves are presented in Figure 4, Figure 5, Figure 6, Figure 7, Appendix A, and the results are summarized in Table 3. Overall, these analyses indicate that the prognostic signature has a good predictive value.

### 3.7. Nomogram Development

To apply the prognostic signature in clinical settings, based on the results of univariate and multivariate Cox proportional hazard regression analyses, nomograms were constructed to predict the PFI and DSS of BC patients at 5 and 10 years. Each risk factor corresponds to a designated point, determined by drawing a line perpendicular to the point’s axis. The sum of the corresponding risk factor points located on the total points represents the probability of 5- and 10-year PFI or DSS, directly leading, straight down, to the 5- and 10-year PFI or DSS axis (Figure 8A,B). The calibration curves demonstrated that the signature possesses high consistencies in nomogram-predicted and actual results when predicting the 5- and 10-year PFI (Figure 8C,D) or DSS (Figure 8E,F) rate of BC patients. Our data suggested that the nomograms for PFI and DSS exhibited a good predictive efficacy in 5- and 10-year PFI and DSS probabilities.

### 3.8. Relevance of the Prognostic Signature in Clinical Decision-Making

Patients were stratified into two groups for the evaluation of the AJCC stage by combining AJCC stages I and II (*N* = 772) into the low-clinical-risk group (marked as C-), and AJCC stage III and IV (*N* = 253) into the high-clinical-risk group (marked as C+) for statistical analysis. Combining the clinical-risk group and gene-risk group (the low-risk group was marked as G-, and the high-risk group was marked as G+), the total patients were classified into the following four subgroups, G-C- (*N* = 557), G-C+ (*N* = 160), G+C- (*N* = 215), G+C+ (*N* = 93). As expected, KM curves showed that patients in the G-C- subgroup had a significantly better prognosis for PFI, DSS, and OS than those in the other subgroups, and the worst was the G+C+ subgroup (PFI, *p* < 0.0001; DSS, *p* < 0.0001; OS, *p* < 0.0001) (Figure 9A–C).

To further evaluate the prognostic signature’s potential as a predictor of response to chemotherapy, KM analysis was performed in the four subgroups. In the G-C- subgroup, the patients who underwent adjuvant chemotherapy had a significantly better prognosis than those who did not (PFI, *p* = 0.073; DSS, *p* = 0.0024; OS, *p* < 0.0001) (Figure 9D–F). In the G-C+ and G+C- subgroup, the patients who underwent adjuvant chemotherapy had a significantly better prognosis for OS than those who did not, but not for PFI and DSS (G-C+, PFI, *p* = 0.34, DSS, *p* = 0.0089, OS, *p* = 0.00019; G+C-, PFI, *p* = 0.17, DSS, *p* = 0.23, OS, *p* = 0.02) (Figure 9G–I,J–L). However, in the G+C+ subgroup, the patients who underwent adjuvant chemotherapy did not show a statistically better prognosis than those who did not (PFI, *p* = 0.2, DSS, *p* = 0.2, OS, *p* = 0.19) (Figure 9M–O). These results suggest that patients in the G-C- subgroup could benefit from adjuvant chemotherapy for PFI, DSS, and OS, while those in the G+C+ subgroup may not, and the patients in the G+C- and G-C+ subgroups could benefit from adjuvant chemotherapy only for OS, not for PFI and DSS.

## 4. Discussion

Breast cancer has become a serious threat to the health of women worldwide; thus, it is imperative to find an effective individualized precision therapy. Although some multigene prognosis tools have been developed to assist in clinical decision-making for patients with BC, the scope of application of these predictors has been somewhat limited; for example, Oncotype Dx and MammaPrint were developed for a specific type and clinical stage of BC [17,18]. Due to the disease heterogeneity, traditional chemotherapy, endocrine therapy, and targeted therapy struggle to achieve effective therapeutic effects for some patients with BC [35]. The role of immunity in BC is becoming clearer; immunotherapy in breast cancer is currently gaining ground, in combination with existing therapeutic strategies [19]. The development of pyroptosis, a highly inflammatory form of programmed cell death, is closely associated with the immune-related functions and infiltration of immune cells in tumors [23,24,25,26]. Therefore, more effort is needed to develop pyroptosis-related prognostic and diagnostic models for BC.

In the present study, two pyroptosis-related clusters were constructed by 38 differential pyroptosis-related genes. Further analysis found that some types of tumor-infiltrating immune cells were variously distributed in different pyroptosis-related clusters, such as naïve B cells, plasma cells, T cells CD8, T cells CD4 memory resting, T cells CD4 memory-activated, T cells follicular helper, T cells gamma delta, NK cells resting, macrophages M0, macrophages M1, macrophages M2, dendritic cells resting, mast cells resting, mast cells activated, and neutrophils. A total of 434 DEGs were identified between the two pyroptosis-related clusters and GO functional enrichment and KEGG pathway analyses demonstrated that the DEGs were enriched in immune-related pathways, such as humoral immune response, adaptive immune response based on somatic immune recombination, B-cell-mediated immunity, complement activation, and immunoglobulin-mediated immune response. These results suggested that pyroptosis is closely related to immunity, which is in accordance with previous studies [36,37].

Based on the DEGs, we strived to identify a 56-gene signature that is significantly associated with the PFI, DSS, and OS of BC. The 56 genes were AC092580.4, AC244250.2, ACKR1, CD1E, CD38, CD48, CD5, CD69, CD79B, CLEC10A, CXCL13, EOMES, GBP5, GPR18, IGHV1-67, IGHV1OR15-2, IGHV2-5, IGHV3-49, IGHV3-64, IGHV3OR16-13, IGHV4-59, IGHV5-51, IGKV1-12, IGKV1-27, IGKV1D-16, IGKV2D-29, IGKV6D-21, IGLC6, IGLC7, IGLV1-44, IGLV2-18, IGLV2-23, IGLV3-1, IGLV7-46, IL18RAP, JAML, LTA, PLAC8, PTGDS, RP11-1094M14.8, RP5-887A10.1, SIRPG, SNX20, SPIB, STAT4, TBC1D10C, TESPA1, THEMIS, TIFAB, TNFRSF17, TRAV12-3, TRAV4, TRBV4-2, TRBV6-6, TRDV1, VPREB3. Some of them are part of the immunoglobulin complex and participate in further humoral immune responses [38,39]; for instance, IGHV1-67, IGHV1OR15-2, IGHV2-5, IGHV3-49, IGHV3-64, IGHV3OR16-13, IGHV4-59, IGHV5-51, IGKV1-12, IGKV1-27, IGKV1D-16, IGKV2D-29, IGKV6D-21, IGLC6, IGLC7, IGLV1-44, IGLV2-18, IGLV2-23, IGLV3-1, IGLV7-46. Some others are part of the receptors and involved in immune response [40,41,42]; for instance, CD1E, CD38, CD48, CD5, CD69, CD79B, CLEC10A, TRAV12-3, TRAV4, TRBV4-2, TRBV6-6, TRDV1.

Further analyses, Kaplan–Meier, ROC analyses, and univariate, multivariate Cox regression demonstrated the utility of this prognostic signature as a powerful predictor of prognosis in patients with BC. A nomogram constructed by combining the prognostic signature and conventional prognostic factors exhibited good predictive efficacy for the prediction of the 5- and 10-year PFI and DSS of patients with BC. Further intensive analyses are required to verify the clinical application and promotion value of the signature. By combining the clinical-risk group and gene-risk group, the patients were classified into the following four subgroups, G-C- (*N* = 557), G-C+ (*N* = 160), G+C- (*N* = 215), G+C+ (*N* = 93). KM analyses suggested that patients in the G-C- subgroup could benefit from adjuvant chemotherapy for PFI, DSS, and OS, while those in the G+C+ subgroup may not, and patients in the G+C- and G-C+ subgroups could benefit from adjuvant chemotherapy for OS, but not for PFI and DSS. Although the patients in the G+C+ subgroup had no significant survival benefit from adjuvant chemotherapy, the adjuvant chemotherapy might improve their quality of life. These results provide theoretical evidence for future clinical decision-making, and further studies are needed to prove the results.

TNBC is characterized by high heterogeneity, high invasion, high metastatic potential, easy recurrence, and poor prognosis [13]. Indeed, great efforts have been dedicated to finding druggable targets for the personalized treatment of TNBC, leading to the discovery of potential and important targets and pathways linked to cancer development [9,43,44], immunity [45] and even pyroptosis [46]. Jiang YZ et al. classified TNBCs into four transcriptome-based subtypes based on the clinical, genomic, and transcriptomic data of a cohort of 465 primary TNBC [47], and the phase Ib/II FUTURE trial suggested a new concept for TNBC treatment, demonstrating the clinical benefit of subtyping-based targeted therapy for refractory metastatic TNBC [48]. Special attention was paid to that in this study. We also found that patients in the low-risk group had significantly better prognoses for PFI, DSS and OS than those in the high-risk group (PFI, *p* < 0.0001; DSS, *p* < 0.0001; OS, *p* < 0.0001). These results provide important indicators for further studies on the therapeutic value of pyroptosis in TNBC.

In the recent studies about BC, four multi-gene signatures (model) have been developed to predict prognosis [49,50,51,52]: a nine pyroptosis-associated lncRNA signature, a seven pyroptosis-related lncRNAs model, three different pyroptosis clusters and a three-gene signature. However, our model has several advantages as a predictor of prognosis in patients with BC. First, our study included 1025 female cases with BC, excluding males and cases with missing clinical information or RNA-seq data, which avoided the possibility of sex-specific effects and ensured more credible results. Second, PFI was chosen as a clinical outcome when constructing a prognostic prediction model, rather than OS, as OS is less sensitive to BC-specific progression. Third, the patient G-C- subgroup in our model could benefit from adjuvant chemotherapy for PFI, DSS, and OS, and patients in the G+C- and G-C+ subgroups could benefit from adjuvant chemotherapy in terms of OS, but not for PFI and DSS. The results could inform clinical decision-making regarding appropriate treatment strategies for patients with BC.

Notwithstanding, the study has a few shortcomings and limitations which should be acknowledged. First, it may take a long time to apply these findings to the clinic in the real world. Although limited by the follow-up time and the number of cases, our Breast Center has already started to establish a validation dataset for BC to verify the findings in this research. If the validation dataset is consistent with the results of this study, the finding will be applied to a prospective clinical study. Second, the biological functions of the 56 genes remain to be fully elucidated. Third, in the M1 subgroup (*N* = 16), the KM curve or ROC subgroup analyses did not reveal any significant difference for DSS and OS, and Luminal B (*N* = 176) and HER2 (*N* = 70) also showed no significant difference for OS. For the M1 subgroup, the number of clinical samples was too small, making it hard to draw a scientific conclusion. Additionally, the heterogeneity of the Luminal B and HER2 molecular subtypes, especially intratumor heterogeneity, presents substantial challenges in cancer treatment; therefore, further studies are needed to identify more accurate molecular models for these patient subgroups. Fourth, our study lacks an independent validation dataset. In the early stage of study design, we considered randomly dividing the full training set into training and validation datasets according to the different proportions, but the larger the sample size in the dataset, the higher the credibility of the established model, so the full dataset was selected for training and modeling. Although the verification dataset was randomly selected from the total dataset with an overlap in the sample points, this also could verify the reliability of the model.

## 5. Conclusions

In conclusion, we identified a novel 56-gene prognostic signature that is significantly associated with PFI, DSS and OS in patients with BC, and developed a nomogram based on this signature with a high prognostic prediction value. Moreover, our data suggested that the prognostic signature, combined with the TNM stage, might be a potential therapeutic strategy for individualized clinical decision-making.

## 6. Risk Calculation Formula

Risk scores = (−0.987057257203668) × AC092580.4 + (−0.251134164354081) × AC244250.2 + (−0.696889063257661) × ACKR1 + (−0.728197493550041) × CD1E + (−1.28880588659078) × CD38 + (−0.749754473910941) × CD48 + 1.06651917547229 × CD5 + 0.417774917378414 × CD69 + 0.455049416172314 × CD79B + 1.26125159638399 × CLEC10A + (−0.228225002214848) × CXCL13 + 0.570516209242267 × EOMES + (−0.548224326281222) × GBP5 + (−0.808563754149334) × GPR18 + 0.309953007309132 × IGHV1-67 + (−0.382414611465128) × IGHV1OR15-2 + 0.246548177641936 × IGHV2-5 + (−0.217409966870419) × IGHV3-49 + (−0.337868541294558) × IGHV3-64 + 0.481005132311589 × IGHV3OR16-13 + 0.362325938526143 × IGHV4-59 + 0.267679621408054 × IGHV5-51 + (−0.206745388340921) × IGKV1-12 + (−0.308185907992354) × IGKV1-27 + (−0.295832726777701) × IGKV1D-16 + 0.176168233864854 × IGKV2D-29 + (−0.180553560947138) × IGKV6D-21 + (−0.353895265830746) × IGLC6 + (−0.13468212251366) × IGLC7 + (−0.223445262734337) × IGLV1-44 + 0.43234569593985 × IGLV2-18 + (−0.261424690345591) × IGLV2-23 + 0.316233011111683 × IGLV3-1 + (−0.198351090291727) × IGLV7-46 + 1.54706517805225 × IL18RAP + (−1.12171432642784) × JAML + (−0.815093088434387) × LTA + 0.529889539295384 × PLAC8 + 0.222843634703247 × PTGDS + 1.03697032621277 × RP11-1094M14.8 + 0.919914395439576 × RP5-887A10.1 + 1.47556877488067 × SIRPG + 2.23000355091989 × SNX20 + 0.306047643348901 × SPIB + (−0.906891717453051) × STAT4 + (−0.734422756261579) × TBC1D10C + (−1.66310277162408) × TESPA1 + 2.21446216265984 × THEMIS + (−1.22937862865569) × TIFAB + 0.884293123405389 × TNFRSF17 + (−0.857408288132703) × TRAV12-3 + 0.719532867205727 × TRAV4 + (−0.62530870110772) × TRBV4-2 + (−0.810480270943384) × TRBV6-6 + (−0.970070159582118) × TRDV1 + (−0.501566873333783) × VPREB3.

## Figures and Tables

**Figure 1 jpm-13-00069-f001:**
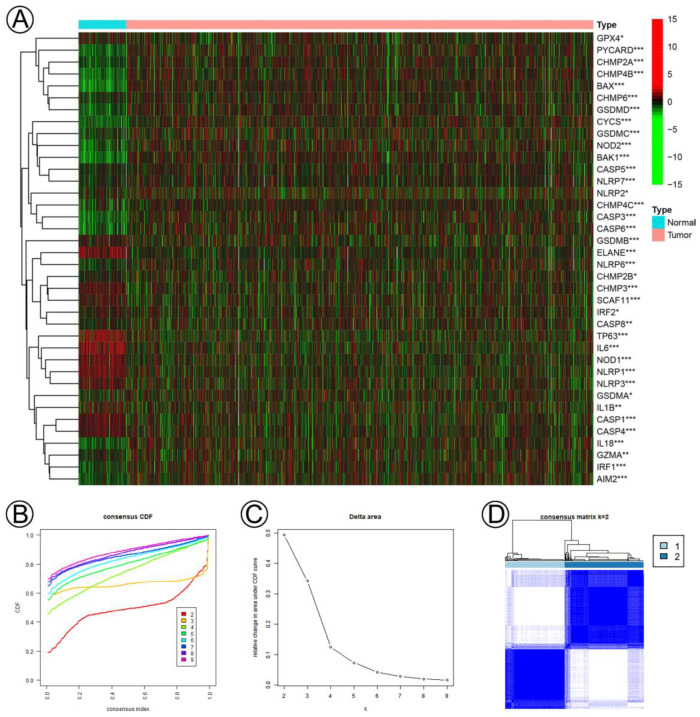
Pyroptosis-related clusters were determined according to 38 pyroptosis-related genes in BC. (**A**) A heatmap of the expression levels of 38 pyroptosis-related genes between the normal and tumor tissues. (**B**,**C**) Consensus clustering CDF and relative change in area under CDF curve for k = 2–9. (**D**) Consensus cluster matrix of breast cancer tumor samples when k = 2. (**E**,**F**) Two-dimensional principal component, and viSNE analysis based on the expression levels of 38 pyroptosis-related genes. The red dots represent C1, and the blue dots represent C2. (**G**–**I**) KM curves of PFI, DSS, and OS for the two pyroptosis-related clusters. Notes: *, *p* < 0.05; **, *p* < 0.01; ***, *p* < 0.001.

**Figure 2 jpm-13-00069-f002:**
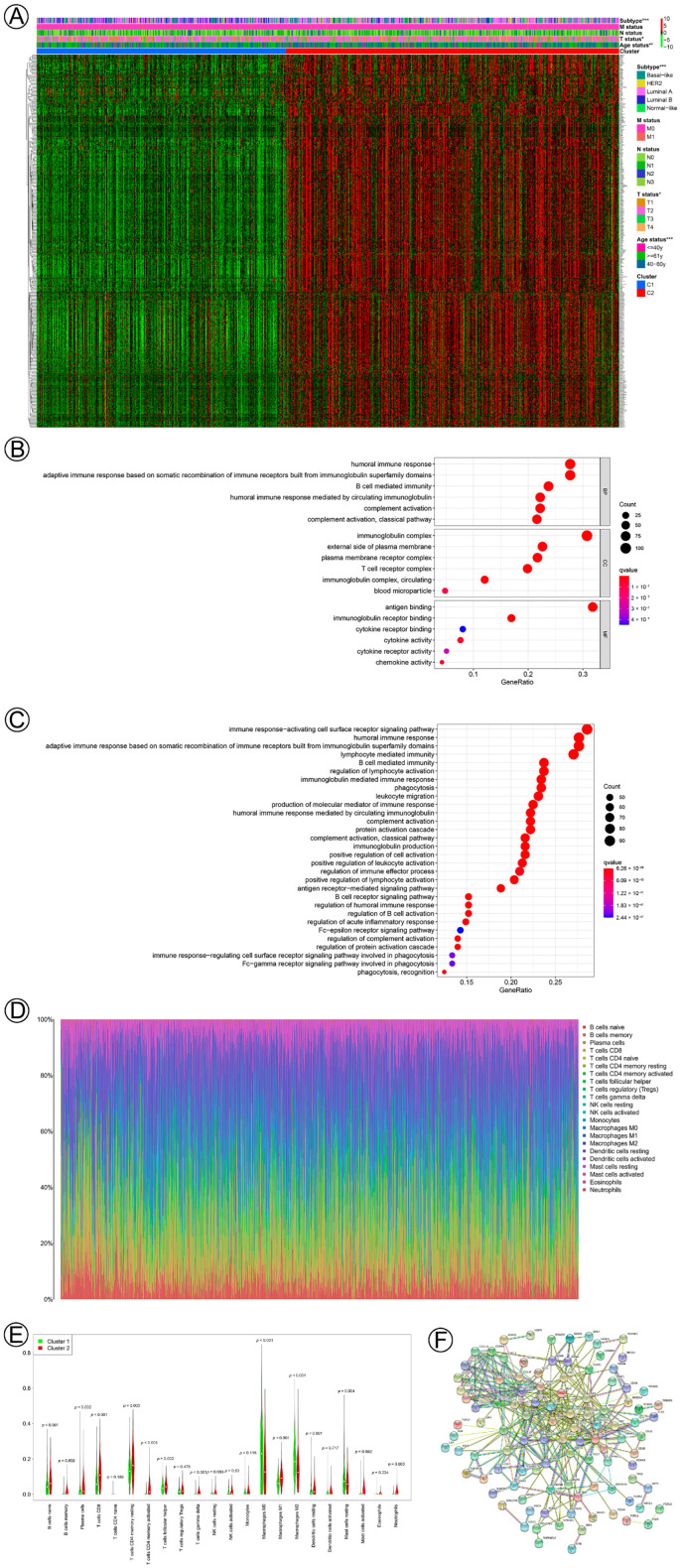
Biological functions and interactions of the differential gene expressions (DGEs) in the two pyroptosis-related clusters in BC. (**A**) A heatmap of the clinical relevance and DGEs (Fold change ≥ 1.5) of the two pyroptosis clusters. (**B**) Gene ontology (GO) annotation (biological process (BP), cellular component (CC), and molecular function (MF) of DGEs. (**C**) KEGG pathway analysis of DGEs. (**D**) Characterization of 22 types of immunocyte infiltration in breast cancer. (**E**) The proportion of immunocyte infiltration in the two pyroptosis-related clusters in breast cancer. (**F**) Protein–protein interaction network of DEGs. Notes: *, *p* < 0.05; **, *p* < 0.01; ***, *p* < 0.001.

**Figure 3 jpm-13-00069-f003:**
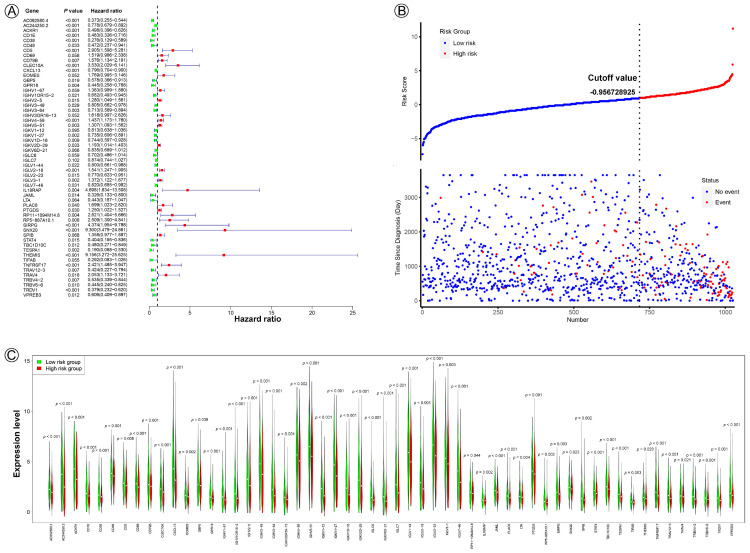
Construction of a 56-gene prognostic signature for BC based on pyroptosis-related DEGs. (**A**) A forest plot of multivariable Cox regression analyses. (**B**) The distribution and survival status of BC patients with different risk scores. The blue and red dots represent clinical events or no clinical events. (**C**) A vioplot of the 56-gene signature in the two risk groups. (**D**–**F**,**J**–**L**), KM curves of PFI, DSS, and OS for high- and low-risk groups in the training and validation datasets. (**G**–**I**,**M**–**O**), ROC analysis shows the sensitivity and specificity of the 56-gene signature for predicting PFI, DSS, and OS for high- and low-risk groups in the training and validation datasets.

**Figure 4 jpm-13-00069-f004:**
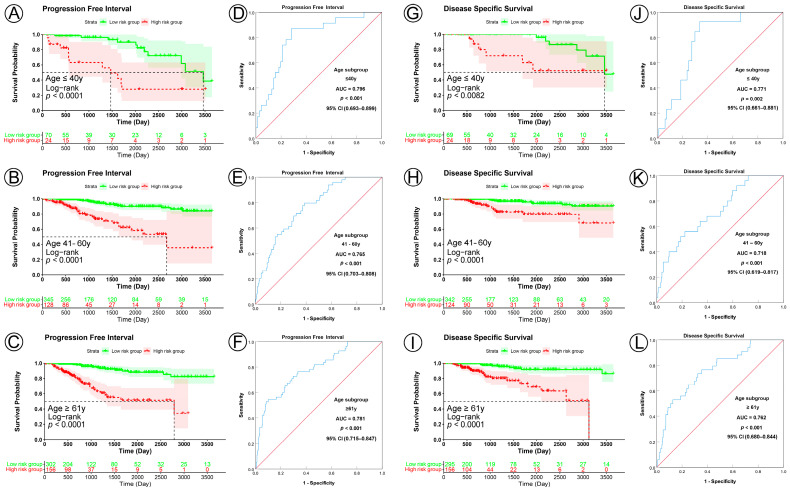
KM and ROC curve analyses of patients stratified by age. (**A**–**C**,**G**–**I**), KM curves of PFI and DSS for high and low-risk groups in the ≤40-year, 41–60-year, and ≥61-year subgroups. (**D**–**F**,**J**–**L**), ROC analysis showed the sensitivity and specificity of prognostic signature for predicting PFI and DSS for high and low-risk groups in the ≤40-year, 41–60-year, and ≥61-year subgroups.

**Figure 5 jpm-13-00069-f005:**
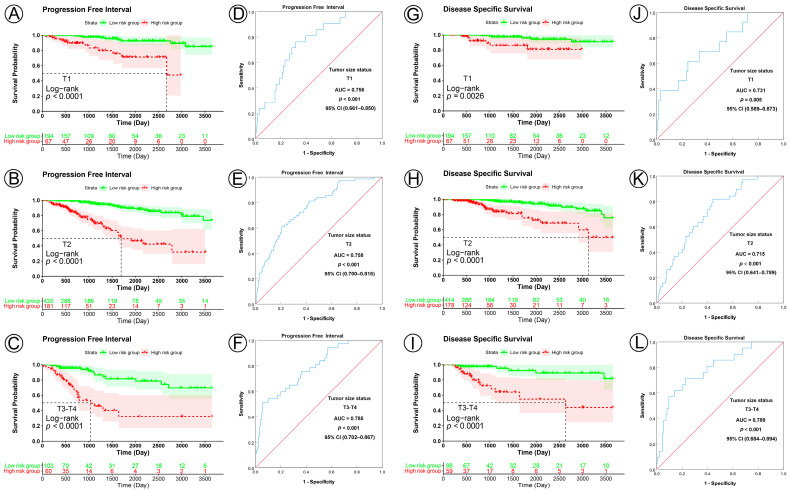
KM and ROC curve analyses of patients stratified by tumor size status. (**A**–**C**,**G**–**I**), KM curves of PFI and DSS for high- and low-risk groups in the T1, T2, and T3–T4 subgroups. (**D**–**F**,**J**–**L**), ROC analysis showed the sensitivity and specificity of prognostic signature for predicting PFI and DSS for high- and low-risk groups in the T1, T2, and T3–T4 subgroups.

**Figure 6 jpm-13-00069-f006:**
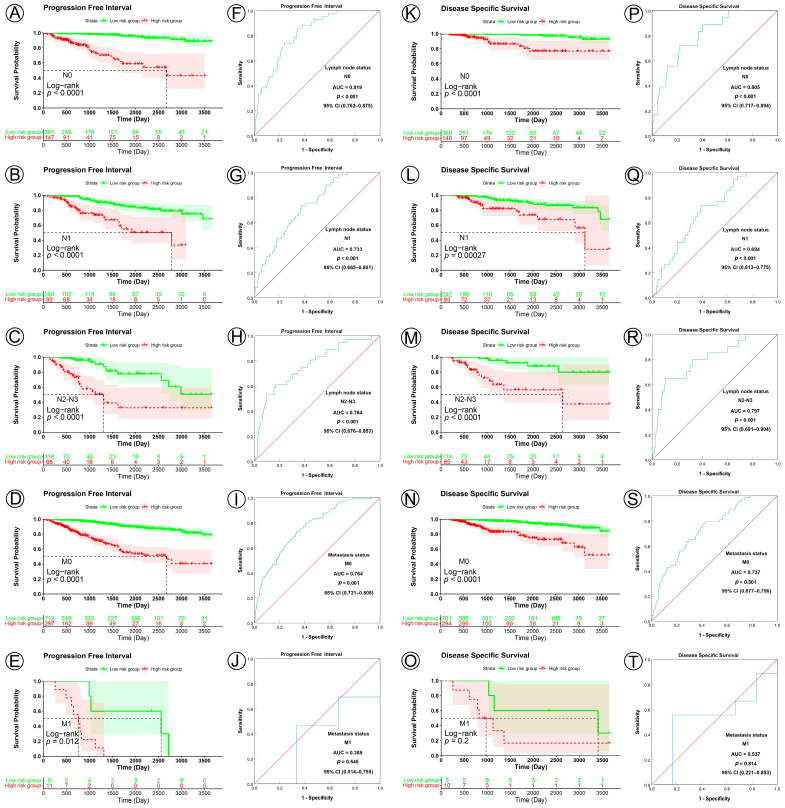
KM and ROC curve analyses of patients stratified by lymph node and metastasis status. (**A**–**E**,**K**–**O**) KM curves of PFI and DSS for high- and low-risk groups in the N0, N1, N2–N3, M0, and M1 subgroups. (**F**–**J**,**P**–**T**) ROC analysis showed the sensitivity and specificity of the prognostic signature for predicting PFI and DSS for high- and low-risk groups in the N0, N1, N2–N3, M0, and M1 subgroups.

**Figure 7 jpm-13-00069-f007:**
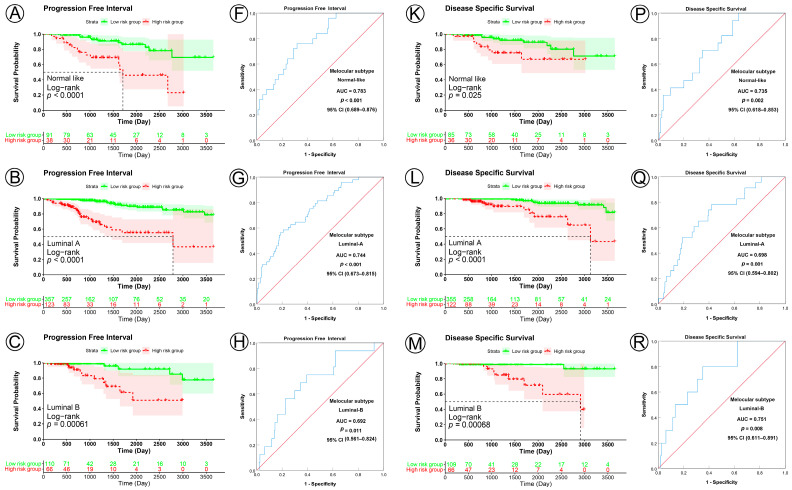
KM and ROC curve analyses of patients stratified by molecular subtype (PAM50). (**A**–**E**,**K**–**O**) KM curves of PFI and DSS for high- and low-risk groups in the normal-like, Luminal A, Luminal B, HER2, Basal-like subgroups. (**F**–**J**,**P**–**T**) ROC analysis showed the sensitivity and specificity of prognostic signature for predicting PFI and DSS for high- and low-risk groups in the normal-like, Luminal A, Luminal B, HER2, Basal-like subgroups.

**Figure 8 jpm-13-00069-f008:**
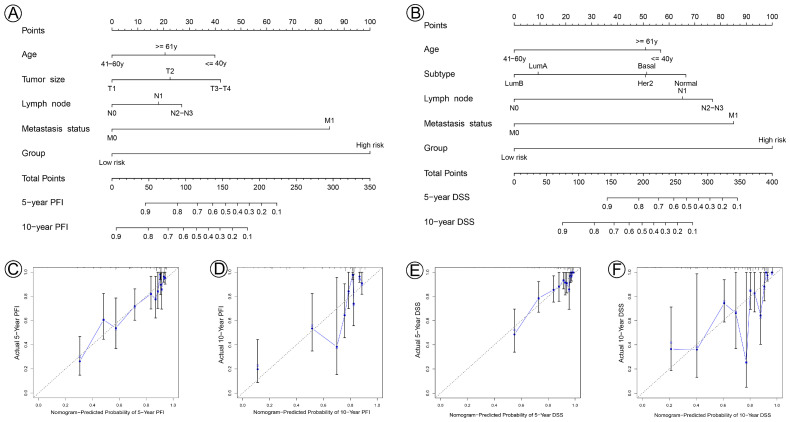
Nomogram for predicting 5- and 10-year PFI and DSS of patients with BC and calibration curves of the nomogram. (**A**) A nomogram incorporating age, tumor size status, lymph node status, metastasis status, and risk group was a predictor for 5- and 10-year PFI. (**B**) A nomogram incorporating age, molecular subtype, lymph node status, metastasis status, and risk group was a predictor for 5- and 10-year DSS. (**C**,**D**) Calibrated plots were applied to investigate the deviation in nomogram-predicted of 5- and 10-year PFI. (**E**,**F**) Calibrated plots were applied to investigate the deviation in nomogram-predicted of 5- and 10-year DSS.

**Figure 9 jpm-13-00069-f009:**
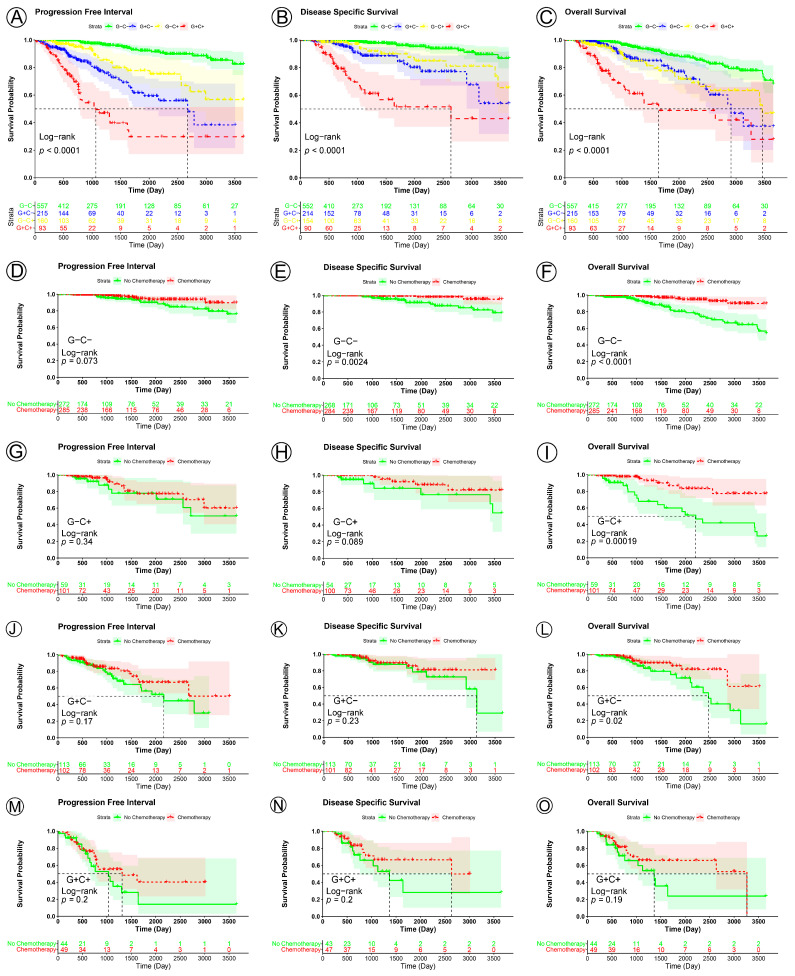
KM curve analyses of patients stratified by four risk subgroups (prognostic signature and clinical risk subgroups). (**A**–**C**), KM curves of PFI, DSS, and OS for breast cancer in the four risk subgroups (G-C-, G-C+, G+C-, and G+C+ subgroups). KM curves of PFI, DSS, and OS for breast cancer in G-C- (**D**–**F**), G-C+ (**G**–**I**), G+C- (**J**–**L**), and G+C+ (**M**–**O**) subgroups.

**Table 1 jpm-13-00069-t001:** Demographic and Clinical, Pathologic Characteristics of Patients with Breast Cancer.

Variable	Training Dataset						Validation Dataset			
	Total	Risk Group		χ^2^	*p* Value	Total	Risk Group		χ^2^	*p* Value
		Low	High				Low	High		
	*n* = 1025	*n* = 717	*n* = 308			*n* = 512	*n* = 355	*n* = 157		
Age, year										
≦40	94	70	24	6.429	0.040	45	33	12	5.661	0.059
41–60	473	345	128			243	179	64		
≧61	458	302	156			224	143	81		
Subtype (PAM50)										
LumA	480	357	123	10.957	0.027	54	35	19	6.174	0.187
LumB	176	110	66			253	186	67		
HER2	70	48	22			86	53	33		
Basal	170	111	59			39	29	10		
Normal	129	91	38			80	52	28		
Tumor size										
T1	261	194	67	5.926	0.052	119	90	29	3.515	0.172
T2	601	420	181			307	210	97		
T3–T4	163	103	60			86	55	31		
Lymph node status										
N0	498	351	147	5.650	0.059	246	170	76	2.033	0.362
N1	343	250	93			168	122	46		
N2–N3	184	116	68			98	63	35		
Metastasis status										
M0	1009	712	297	11.582	0.001	502	352	150	7.243	0.011 ^a^
M1	16	5	11			10	3	7		

^a^. Fisher’s Exact Test.

**Table 2 jpm-13-00069-t002:** Univariate and Multivariate Cox proportional hazard models of PFI and DSS in Breast Cancer.

Variables	Progression-Free Interval						Disease-Specific Survival					
	Univariate			Multivariate			Univariate			Multivariate		
	HR	95% CI	*p* Value	HR	95% CI	*p* Value	HR	95% CI	*p* Value	HR	95% CI	*p* Value
Age												
41–60 year	0.468	0.285–0.769	0.003	0.5014	0.299–0.841	0.009	0.446	0.228–0.439	0.018	0.445	0.217–0.912	0.027
≧61 year	0.686	0.421–1.119	0.131	0.716	0.431–1.191	0.198	0.834	0.439–1.585	0.580	0.972	0.494–1.912	0.935
Subtype (PAM50)												
Luminal-A	0.675	0.416–1.095	0.112				0.417	0.222–0.783	0.006	0.414	0.218–0.787	0.007
Luminal-B	0.694	0.370–1.301	0.255				0.589	0.269–1.289	0.185	0.378	0.168–0.851	0.019
HER2	1.040	0.485–2.230	0.920				0.757	0.279–2.054	0.584	0.810	0.291–2.258	0.687
Basal-like	1.217	0.712–2.081	0.473				0.910	0.464–1.787	0.785	0.841	0.415–1.706	0.632
Tumor size												
T2	1.865	1.144–3.042	0.013	1.476	0.891–2.447	0.131	1.656	0.881–3.110	0.117	0.994	0.508–1.947	0.986
T3–T4	3.643	2.131–6.228	<0.001	2.073	1.145–3.751	0.016	3.126	1.561–6.257	0.001	1.483	0.682–3.228	0.320
Lymph node status												
N1	1.670	1.106–2.522	0.015	1.367	0.893–2.094	0.150	2.723	1.538–4.822	0.001	2.592	1.421–4.730	0.002
N2–N3	3.151	2.015–4.929	<0.001	1.597	0.953–2.677	0.075	4.137	2.186–7.830	<0.001	2.767	1.327–5.773	0.007
Metastasis status												
M1	7.804	4.386–13.900	<0.001	4.305	2.261–8.194	<0.001	7.053	3.489–14.260	<0.001	3.553	1.617–7.807	0.002
Risk group												
High-risk	6.257	4.331–9.039	<0.001	5.643	3.894–8.175	<0.001	5.520	3.407–8.944	<0.001	4.578	2.797–7.494	<0.001

PFI, Progression-Free Interval; CI, Confidence Interval; HR, Hazard Ratio.

**Table 3 jpm-13-00069-t003:** Result of Kaplan–Meier and ROC analysis based on different regrouping methods.

			Progression-Free Interval				Disease-Specific Survival			
Regrouping Factors	Subgroup	Sample Size	Kaplan–Meier	ROC			Kaplan–Meier	ROC		
			*p* Value	AUC	95% CI	*p* Value	*p* Value	AUC	95% CI	*p* Value
Age, y										
	≦40	94	<0.001	0.796	0.693–0.899	<0.001	0.001	0.771	0.661–0.881	0.002
	41–60	473	<0.001	0.765	0.703–0.808	<0.001	<0.001	0.718	0.619–0.817	<0.001
	≧61	458	<0.001	0.781	0.715–0.847	<0.001	<0.001	0.762	0.680–0.844	<0.001
Tumor size status										
	T1	261	<0.001	0.756	0.661–0.850	<0.001	0.003	0.731	0.589–0.873	0.005
	T2	601	<0.001	0.758	0.700–0.816	<0.001	<0.001	0.715	0.641–0.789	<0.001
	T3–T4	163	<0.001	0.785	0.702–0.867	<0.001	<0.001	0.789	0.648–0.894	<0.001
Lymph node status										
	N0	498	<0.001	0.819	0.762–0.875	<0.001	<0.001	0.805	0.717–0.894	<0.001
	N1	343	<0.001	0.733	0.665–0.801	<0.001	<0.001	0.694	0.613–0.775	<0.001
	N2–N3	184	<0.001	0.764	0.676–0.853	<0.001	<0.001	0.797	0.691–0.904	<0.001
Metastasis status										
	M0	1009	<0.001	0.764	0.721–0.808	<0.001	<0.001	0.737	0.677–0.796	<0.001
	M1	16	0.012	0.385	0.014–0.755	0.545	0.200	0.537	0.221–0.853	0.814
Subtype (PAM50)										
	Normal like	129	<0.001	0.783	0.689–0.876	<0.001	0.025	0.735	0.618–0.853	0.002
	Luminal-A	480	<0.001	0.744	0.673–0.815	<0.001	<0.001	0.698	0.594–0.802	0.001
	Luminal-B	176	<0.001	0.692	0.561–0.824	0.011	<0.001	0.751	0.611–0.891	0.008
	HER2	70	<0.001	0.854	0.753–0.955	0.001	0.006	0.851	0.712–0.989	0.009
	Basal like	170	<0.001	0.813	0.738–0.888	<0.001	<0.001	0.789	0.684–0.894	<0.001

ROC, receiver operating characteristic; AUC, area under the curve; CI, confidence interval.

## Data Availability

This study was performed using data from the Cancer Genome Atlas Research Network, and the data can be obtained through the TCGA Genomic Data Commons portal (GDC, https://cancergenome.nih.gov/ (Data Release 31.0—29 October 2021)).

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
