# Peer review of "A Novel Prognostic Prediction Model Based on Pyroptosis-Related Clusters for Breast Cancer"

_jpm, 2022, doi:10.3390/jpm13010069_

Round 1
Reviewer 1 Report
The study is well done, the material is large enough and the methods look reliable. The study is based on extensive and not recent literature, gives some new information and this warrants its publication.
Although several reviews about the diagnostics of breast cancer have been already published, the discussion on the markers of breast carcinoma in this paper seems to be original. However I have the following suggestions/comments and hope the authors can address them in the review.
Minor revision
1. Some authors showed that the ADH/ALDH activities are lower in tumor cells than in normal parenchyma, suggesting that isoenzymes of ADH may play an important role in carcinogenesis. Among all tested classes of ADH isoenzymes, only class I had higher activity in the serum of patients with breast cancer in stage IV:
Jelski W et al.,
The activity of class I, II, III and IV alcohol dehydrogenase isoenzymes and aldehyde dehydrogenase in breast cancer. Clin Exp Med (2006) 6:89–93
Jelski W et al.,
Activity of Alcohol Dehydrogenase (ADH) Isoenzymes and Aldehyde Dehydrogenase (ALDH) in the Sera of Patients with Breast Cancer.
Journal of (5-6 sentence)
Author Response
The study is well done, the material is large enough and the methods look reliable. The study is based on extensive and not recent literature, gives some new information and this warrants its publication.
Although several reviews about the diagnostics of breast cancer have been already published, the discussion on the markers of breast carcinoma in this paper seems to be original. However I have the following suggestions/comments and hope the authors can address them in the review.
Minor revision
1.Some authors showed that the ADH/ALDH activities are lower in tumor cells than in normal parenchyma, suggesting that isoenzymes of ADH may play an important role in carcinogenesis. Among all tested classes of ADH isoenzymes, only class I had higher activity in the serum of patients with breast cancer in stage IV:
Thank you for your comments. We have amended the statement as follows (Page 3, line 11-14):
Moreover, some authors showed that the ADH/ALDH activities are lower in tumor cells than in normal parenchyma, suggesting that isoenzymes of ADH may play an important role in carcinogenesis. Additionally, among all tested classes of ADH isoenzymes, only class I had higher activity in the serum of patients with breast cancer in stage IV[14; 15].
Reference
[14] W. Jelski, L. Chrostek, M. Szmitkowski, and W. Markiewicz, The activity of class I, II, III and IV alcohol dehydrogenase isoenzymes and aldehyde dehydrogenase in breast cancer. Clin Exp Med 6 (2006) 89-93
[15] W. Jelski, L. Chrostek, W. Markiewicz, and M. Szmitkowski, Activity of alcohol dehydrogenase (ADH) isoenzymes and aldehyde dehydrogenase (ALDH) in the sera of patients with breast cancer. J Clin Lab Anal 20 (2006) 105-108
Reviewer 2 Report
Bao-Xing Tian et al. manuscript proposes a novel prognostic gene signature associated with progression free interval, disease specific survival and overall survival in breast cancer patients. The proposed novel prognostic signature, consisting of 56 genes identified via RNA-seq and TGCA analysis, is based on pyroptosis-related clusters and, combined with TNM staging, is here proposed to as a potential strategy to route and suggest the therapeutic strategy for high-valued personalised approach and individualised clinical decision-making. Considering the recognised role of pyroptosis in tumour development and progression and the importance of the correlated immune response in enhancing the effectiveness of guide-line-approved therapies, the topic covered by the authors is timely and of great interest for the research field. While the abstract is correctly structured and provides a comprehensive summary of the authors’ work and aim, some adjustments and corrections are required.
This reviewer suggests acceptance of this work with these major and minor revisions required for publication suitability:
1) Authors are invited to submit their manuscript for proofreading check by an English native speaker for correct syntax to ensure a complete understanding of their work (e.g. Introduction section, line 5 “…which based on tumor hormonal status…”. Punctuation requires an in-depth check as well (e.g. Introduction, line 7, “…Ki67 status [2;3]. And the patient…”. Short colloquial forms must be avoided (e.g. Introduction, line 11, “But, it’s…”) and the long formal form should be preferred.
2) The Introduction section lacks pivotal information, since the authors only partly reported the state of the art of interest for their work. Based on TNM staging, breast cancer is sub-divided into five different stages (0, I, II, III, IV) and not just early and advanced breast cancer. This is of pivotal interest also considering that the correct therapy is chosen also depending on the stage. In this regard, the therapeutic landscape for the different breast cancer subtypes should be deepened and the use of radiotherapy mentioned. Finally, the importance of pyroptosis in cancer development is not contextualised and a clear and detailed explanation of why this mechanism is of pivotal importance and interest for the field is lacking. The additions and corrections here suggested would definitely improve the quality of this work and help to better convey the authors’ message and findings.
3) Figures labelling must follow the order in which they are presented in the results section, e.g. Figure 2F cannot be cited before Figure 2B. Please correct the figure order or correct the results presentation.
4) The addition of the authors’ considerations and point of view on how TNBC treatment could benefit from their study and how this prognostic signature could influence therapy and clinical decision-making of TNBC would definitely and greatly improve the discussion part due to the still unmet medical need represented by TNBC. Indeed, great effort has been made to find druggable targets for the personalised treatment of TNBC, leading to the discovery of potential and important targets and pathways linked to cancer development (e.g. OXER1 and RACK1, Masi M et al. Oncogenesis. 2020 Dec 11;9(12):105. doi: 10.1038/s41389-020-00291-x; Masi M et al. Cells. 2021 Nov 3;10(11):2999. doi: 10.3390/cells10112999; Buoso E et al. Br J Pharmacol. 2022 Jun;179(12):2813-2828. doi: 10.1111/bph.15218), immunity (e.g. RACK1, Corsini E et al. Adv Exp Med Biol. 2021;1275:151-163. doi: 10.1007/978-3-030-49844-3_6) and even pyroptosis (e.g. RACK1 and NLRP3, Wei X et al. Cell Mol Immunol. 2022 Sep;19(9):971-992. doi: 10.1038/s41423-022-00905-x).
Author Response
Review2:
Bao-Xing Tian et al. manuscript proposes a novel prognostic gene signature associated with progression free interval, disease specific survival and overall survival in breast cancer patients. The proposed novel prognostic signature, consisting of 56 genes identified via RNA-seq and TGCA analysis, is based on pyroptosis-related clusters and, combined with TNM staging, is here proposed to as a potential strategy to route and suggest the therapeutic strategy for high-valued personalised approach and individualised clinical decision-making. Considering the recognised role of pyroptosis in tumour development and progression and the importance of the correlated immune response in enhancing the effectiveness of guide-line-approved therapies, the topic covered by the authors is timely and of great interest for the research field. While the abstract is correctly structured and provides a comprehensive summary of the authors’ work and aim, some adjustments and corrections are required.
This reviewer suggests acceptance of this work with these major and minor revisions required for publication suitability:
1)Authors are invited to submit their manuscript for proofreading check by an English native speaker for correct syntax to ensure a complete understanding of their work (e.g. Introduction section, line 5 “…which based on tumor hormonal status…”. Punctuation requires an in-depth check as well (e.g. Introduction, line 7, “…Ki67 status [2;3]. And the patient…”. Short colloquial forms must be avoided (e.g. Introduction, line 11, “But, it’s…”) and the long formal form should be preferred.
We appreciate the comments. We have used the recommended editing services to polish the English of the manuscript (English-edited No. 56946), and improved the English in the revised manuscript.
2)The Introduction section lacks pivotal information, since the authors only partly reported the state of the art of interest for their work. Based on TNM staging, breast cancer is sub-divided into five different stages (0, I, II, III, IV) and not just early and advanced breast cancer. This is of pivotal interest also considering that the correct therapy is chosen also depending on the stage. In this regard, the therapeutic landscape for the different breast cancer subtypes should be deepened and the use of radiotherapy mentioned. Finally, the importance of pyroptosis in cancer development is not contextualised and a clear and detailed explanation of why this mechanism is of pivotal importance and interest for the field is lacking. The additions and corrections here suggested would definitely improve the quality of this work and help to better convey the authors’ message and findings.
Thank your advice. After careful discussion, we agreed with the reviewer's opinions and useful constructive opinions, and agreed that this statement was not completely accurate. Therefore, we have amended the statement as follows (Page 3, line 7-9):
Based on TNM staging, breast cancer was categorized into five different stages (0, I, II, III, IV)[4]. In the course of clinical personalized treatment, patients are further sub-divided into advanced-stage (stage III and IV) and early-stage (stage 0, I and II) BC[5; 6].
3)Figures labelling must follow the order in which they are presented in the results section, e.g. Figure 2F cannot be cited before Figure 2B. Please correct the figure order or correct the results presentation.
Thank your comment. We had adjusted the presentation of the result in order (Page 7, line 128-129).
4)The addition of the authors’ considerations and point of view on how TNBC treatment could benefit from their study and how this prognostic signature could influence therapy and clinical decision-making of TNBC would definitely and greatly improve the discussion part due to the still unmet medical need represented by TNBC. Indeed, great effort has been made to find druggable targets for the personalised treatment of TNBC, leading to the discovery of potential and important targets and pathways linked to cancer development (e.g. OXER1 and RACK1, Masi M et al. Oncogenesis. 2020 Dec 11;9(12):105. doi: 10.1038/s41389-020-00291-x; Masi M et al. Cells. 2021 Nov 3;10(11):2999. doi: 10.3390/cells10112999; Buoso E et al. Br J Pharmacol. 2022 Jun;179(12):2813-2828. doi: 10.1111/bph.15218),immunity (e.g. RACK1, Corsini E et al. Adv Exp Med Biol. 2021;1275:151-163. doi: 10.1007/978-3-030-49844-3_6) and even pyroptosis (e.g. RACK1 and NLRP3, Wei X et al. Cell Mol Immunol. 2022 Sep;19(9):971-992. doi: 10.1038/s41423-022-00905-x).
Thank your advice. we have amended the statement as follows (Page 22, line 406-415):
TNBC is characterized by high heterogeneity, high invasion, high metastatic potential, easy recurrence, and poor prognosis[13]. Indeed, great efforts have been dedicated to finding druggable targets for the personalized treatment of TNBC, leading to the discovery of potential and important targets and pathways linked to cancer development[9; 43; 44], immunity[45] and even pyroptosis[46]. Jiang YZ et al classified TNBCs into four transcriptome-based subtypes based on the clinical, genomic, and transcriptomic data of a cohort of 465 primary TNBC[47], and the phase Ib/II FUTURE trial suggested a new concept for TNBC treatment, demonstrating the clinical benefit of subtyping-based targeted therapy for refractory metastatic TNBC[48]. Special attention was paid to that in this study. We also found that patients in the low-risk group had significantly better prognosis for PFI, DSS and OS than those in the high-risk group (PFI, P < 0.0001; DSS, P < 0.0001; OS, P < 0.0001). These results provide important indicators for further studies on the therapeutic value of pyroptosis in TNBC.
Reviewer 3 Report
In this study, the authors constructed a novel prognostic prediction model for breast cancer, based on pyroptosis-related clusters according to RNA-seq and clinical data downloaded from TCGA. A 56 genes signature constructed by using univariate and multivariate Cox regression, was significantly associated with progression free interval (PFI), disease specific survival (DSS), and overall survival (OS) of patients with BC. Cox analysis revealed that the signature was significantly associated with PFI and DSS of patients. The authors use this signature to distinguish high and low risk patients and exhibited high sensitivity and specificity in predicting the prognosis. Combining with clinical risk, patients in both gene and clinical low risk subgroup who received adjuvant chemotherapy had a significantly lower incidence of the clinical event than those who did not.
Major comments:
1. The 56 gene signature was generated from the 1025 tumor patients. But the authors randomly selected 512 patients from the original 1025 patients as the validation group. The validation set SHOULD be independent.
2. What is the use of the 434 DEGs? Seems have nothing to do with the predictive signature.
Minor comment:
3. The manuscript needs to be further edited for language.
Author Response
Review3:
In this study, the authors constructed a novel prognostic prediction model for breast cancer, based on pyroptosis-related clusters according to RNA-seq and clinical data downloaded from TCGA. A 56 genes signature constructed by using univariate and multivariate Cox regression, was significantly associated with progression free interval (PFI), disease specific survival (DSS), and overall survival (OS) of patients with BC. Cox analysis revealed that the signature was significantly associated with PFI and DSS of patients. The authors use this signature to distinguish high and low risk patients and exhibited high sensitivity and specificity in predicting the prognosis. Combining with clinical risk, patients in both gene and clinical low risk subgroup who received adjuvant chemotherapy had a significantly lower incidence of the clinical event than those who did not.
Major comments:
1.The 56 gene signature was generated from the 1025 tumor patients. But the authors randomly selected 512 patients from the original 1025 patients as the validation group. The validation set SHOULD be independent.
Thanks for your comment. In the early stage of study design, we had considered that the full dataset was randomly divided into training and validation datasets by the proportion of 1:1, 3:2, 7:3, 4:1, but the larger the sample size of the dataset, the higher the credibility of the model established, so absolutely select full dataset for training and modeling. Although the verification dataset was randomly selected from the total dataset with the overlap of the sample points, it can also verify the reliability of the model.
2.What is the use of the 434 DEGs? Seems have nothing to do with the predictive signature.
Thanks for your question. A total of 434 DEGs were identified by Wilcoxon test based on the two pyroptosis-related clusters, and shown in the pheatmap (Figure 2 A). Then, the further bioinformatic analysis of the two pyroptosis-related clusters were based on the 434 DEGs, and the KEGG pathways and GO terms demonstrated that the DEGs were enriched in immune-related pathways. Furthermore, the 434 DEGs were used to seek prognostic value by univariate Cox proportional hazard analysis at the single gene level.
Minor comment:
3.The manuscript needs to be further edited for language.
We appreciate the comments. We have used the recommended editing services to polish the English of the manuscript (English-edited No. 56946), and have improved the English in the revised manuscript.
